# Fractal Kinetic Implementation in Population Pharmacokinetic Modeling

**DOI:** 10.3390/pharmaceutics15010304

**Published:** 2023-01-16

**Authors:** Woojin Jung, Hyo-jeong Ryu, Jung-woo Chae, Hwi-yeol Yun

**Affiliations:** 1College of Pharmacy, Chungnam National University, Daejeon 34134, Republic of Korea; 2Department of Bio-AI Convergence, Chungnam National University, Daejeon 34134, Republic of Korea

**Keywords:** pharmacokinetics, compartment modeling, fractal, transdermal patch, intramuscular injection, biological drug

## Abstract

Compartment modeling is a widely accepted technique in the field of pharmacokinetic analysis. However, conventional compartment modeling is performed under a homogeneity assumption that is not a naturally occurring condition. Since the assumption lacks physiological considerations, the respective modeling approach has been questioned, as novel drugs are increasingly characterized by physiological or physical features. Alternative approaches have focused on fractal kinetics, but evaluations of their application are lacking. Thus, in this study, a simulation was performed to identify desirable fractal-kinetics applications in conventional modeling. Visible changes in the profiles were then investigated. Five cases of finalized population models were collected for implementation. For model diagnosis, the objective function value (OFV), Akaike’s information criterion (AIC), and corrected Akaike’s information criterion (AICc) were used as performance metrics, and the goodness of fit (GOF), visual predictive check (VPC), and normalized prediction distribution error (NPDE) were used as visual diagnostics. In most cases, model performance was enhanced by the fractal rate, as shown in a simulation study. The necessary parameters of the fractal rate in the model varied and were successfully estimated between 0 and 1. GOF, VPC, and NPDE diagnostics show that models with the fractal rate described the data well and were robust. In the simulation study, the fractal absorption process was, therefore, chosen for testing. In the estimation study, the rate application yielded improved performance and good prediction–observation agreement in early sampling points, and did not cause a large shift in the original estimation results. Thus, the fractal rate yielded explainable parameters by setting only the heterogeneity exponent, which reflects true physiological behavior well. This approach can be expected to provide useful insights in pharmacological decision making.

## 1. Introduction

In the field of pharmacokinetic (PK) analysis, compartmental modeling is a widely accepted technique that is used in the various clinical stages of drug development [1]. In this approach, an ordinary differential equation (ODE) is commonly used to describe the quantitative relationships between compartments in both simple and complex structured models. ODEs are used to represent the changes in the drug amount in a given compartment for a specified period, with the amount of change expressed as the ordered rate, described as a fixed parameter over time on the basis of a homogeneity assumption; however, this condition rarely occurs in natural processes [2]. Consequently, conventional modeling based on a homogeneity assumption does not consider the actual space in which drug diffusion and transportation take place; instead, the characteristics of the space are either ignored or abstracted into simple processes during the PK analysis.

Since all drug dosing is necessarily carried out on the basis of diffusion towards the site of action through distinct kinds of interfaces, the physical features of the drug body (in other words, the drug depot) and the physiological features of the drug administration site play important roles in determining the PK. The homogeneity assumed in conventional methods does not adequately address these issues, such that, in the PK profile, it is difficult to clearly distinguish among absorption, distribution, metabolism and excretion (ADME).

For example, under clinical conditions, transdermal-patch drugs that follow zero-ordered rate release patterns in drug studies may not achieve the intended plasma concentration [3], but rather fluctuate over time because the gradient of driving forces that moves the drug molecules out of the formulation and into the central body system is not constant. In this case, during conventional modeling, the degree of freedom for a molecule-taking path is considered to be 1, in which a projectile linearly extends from the formulation to the central system according to Fick’s law. Rather, under physiological conditions, a portion of the drug molecules take the fastest route to the central system, while the remainder are trapped in the formulation. Thus, Fick-type diffusion does not describe the quantitative movement relationship for a degree of freedom > 1, and the respective PK model insufficiently reflects the mechanistic action of the drug.

Recent advances resulting in novel drug molecules and their formulations have highlighted the deficiencies of the conventional approach, as new generations of drugs more closely utilize physical and physiological conditions, such that the models required to describe their PK profiles have become more complicated. An alternative modeling method that can be used to represent drug movement uses ordered rates that vary over time using fractal kinetics, as demonstrated in the study by Kopelman et al. [4,5] In that study, fractal-like kinetics were introduced to describe the rate of reaction in fractal dimension in comparison with classical reaction kinetics. This fractal-like reaction rendered the rate dependent on its elapsed time and enabled the macroscopic interpretation of the response of fractal dimensions (Figure 1). The method was suitable for explaining the dimensional characteristics of molecules in various tissues and organs in several studies.

Dokoumetzidis, and Macheras et al. organized and applied fractal kinetic concepts to PK and pharmacodynamics [6,7]. Other researchers employed fractal kinetics to in vitro in vivo correlation (IVIVC) studies to quantify drug dissolution and release at interfaces [8,9]. Fractal kinetics have been vigorously applied In PK studies of ADME. For example, a fractal dimension was introduced in drug metabolism in the liver using a physiological pharmacokinetic model [10] and in modeling tissue trapped in a distribution space [11,12,13], drug elimination [14]. In those studies, the observed PK profiles were described well using fractal kinetic models, and the concept of the fractal was mechanistically suitable.

Fractal kinetics in PK analysis has been successfully introduced into case-specific studies, but generalized applications to PK analysis have not been investigated through direct comparison with the conventional method in terms of validity and applicability. Thus, in this study, a simple fractal-kinetics expression obtained in previous research and the inclusion of a heterogeneity exponent were implemented together with a conventional population PK model to determine its sensitivity to the model; the fractal expression was then applied to five developed population PK models to assess its performance metrics and diagnostics, and thus confirm the validity and practicality of fractal-kinetics interpretation in the field of PK analysis.

## 2. Materials and Methods

### 2.1. Fractal Rate Expression

The fractal expression introduced by Kopelman et al. [4] was implemented in previously constructed mixed-effect models. In the fractional dimension, the rate constant was divided by time to the power heterogeneity. The heterogeneity was set to be 0–1. The equation is, therefore, as follows:(1)$$rate=\frac{\theta }{time^{h}}$$
where *rate* is the fractal-like rate that is an instantaneous rate coefficient [5], θ is the rate coefficient at the time point of 1, *time* is a modeled time from the point of dosing, *h* is the heterogeneity exponent, and the rate coefficient is regarded to be the rate constant in Fick’s law when *h* is set to 0. In Equation (Equation 1), the term containing ’time’ is considered to be dimensionless, and the scale of time that the ordinary differential equation system solves was put into the modeled time.

### 2.2. Simulation Study

Simulations were conducted using one- and two-compartment models, and fractal expressions were applied on the absorption rate, intercompartmental rates, and elimination rate by replacing the existing rate constant with the instantaneous rate coefficient from Equation (Equation 1). In the one-compartment model, rate constants Ka (absorption rate) and Ke (elimination rate) were modified as fractal expressions. For the two-compartment model, Ka, Ke, rate central to peripheral (Kcp), and rate peripheral to central (Kpc) were modified, and the heterogeneity exponent h was changed by 0.05 within the range of 0–1. The simulated PK profiles covered 0–400 hours, with 100 mcg doses given via the extravascular route (i.e., not administered directly into the central system). The parameters of Ka and CL (clearance, defined as the multiplication of Ke by the central volume of distribution) were set as lower (0.033), middle (0.1), and upper (0.3) values; V2 (the volume of distribution in the peripheral compartment) was set as lower (1) and upper (100) values. These parameters were compared in each possible combination where the rates between compartments were differently assigned. The simulated parameters are summarized in Table 1.

On the basis of simulation research, a fractal rate equation that could be straightforwardly interpreted was selected and applied to several original models constructed on the assumption of a homogeneous environment while maintaining the structures of the models.

### 2.3. Model and Dataset Collection for Real Case Application

Five conventional models were tested. The model parameters were modified using the fractal expressions considered to be the most appropriate for use with the conventional method. The five models were as follows. Single-dose transdermal patch [3]. This was a two-compartment model with a first-order absorption in the case of an oral dose, and absorption in two transit compartments in the case of a transdermal dose. The amounts derived from oral administration and patch administration were processed in the same central compartment. The model was fitted simultaneously in each case of oral and transdermal patch administration for 312 h. Single oral and transdermal-patch amounts were dosed.Multiple doses of a drug administered either orally or via a transdermal patch. This two-compartment model included first-order absorption for the oral dose, and absorption in a transit compartment for the transdermal dose. The drug amounts from oral administration and patch administration were processed in the same central compartment. The model was fitted under the condition of 7 days of titration with the oral dose and then three patch doses, with an observation time of 2496 h. Two different amounts were dosed for oral and transdermal-patch administration.At every point of new dosing, the time value in the equation was modeled to be reset as zero, and the remaining amount was emptied.Controlled-release intramuscular injection (inhouse data). The two-compartment model included ordinary first-order rate absorption and transit absorption using Stirling’s approximation (Savic et al. [15]) to the same central compartment. The dose was divided into two fractions, one with fast absorption and the other with slow absorption. The intramuscular injection was administered once, and the model was fitted for a period of 672 h. Four different drug amounts were dosed.Subcutaneous injection, antibody [16]. This was a two-compartment target-mediated drug disposition (TMDD) model with quasiequilibrium conditions. It consisted of a drug depot (injection site), distribution space, and central and peripheral compartments. The drug concentration was observed for a maximum of 746 h. Five different amounts for subcutaneous injections were dosed.Subcutaneous injection, antibody (anakinra) [16]. This was a one-compartment target-mediated drug disposition (TMDD) model with quasiequilibrium conditions. It consisted of a drug depot (injection site) and a central compartment. The drug concentration was observed for a maximum of 48 h. One amount for subcutaneous injection was dosed.

### 2.4. Model Evaluation

As this was a numerical diagnostics study for the purpose of estimation, model performance was measured using objective function value (OFV), Akaike’s information criterion (AIC) [17], and corrected Akaike’s information criterion (AICc) [18]. The fractal models were compared with the original models with respect to *OFV*, *AIC*, and *AICc*.

*OFV* is defined as minus twice the log of the likelihood. It yields a single number that provides overall model fitting on the basis of the distribution of the observations. *AIC* and *AICc* are defined by the *OFV* value, the number of free parameters, and the number of observations in the model. Decreasing values of these measures indicate better models. The equations are as follows:(2)AIC=OFV+2k
(3)$$AICc=AIC+\frac{2k(k+1)}{n-k-1}$$

A change is regarded significant if the *OFV* decreases to 3.84 for one parameter addition, and to 5.99 for the addition of two parameters. For *AIC* and *AICc*, a decrease in the value is regarded as significant. In addition to numerical diagnostics, visual diagnostics such as the goodness of fit (GOF) and prediction-corrected visual predictive check (pcVPC) [19] were performed for each model. For GOF, observations were compared to population predictions and individual predictions; individual weighted residuals were compared to individual predictions and conditional weighted residuals [20] over the observed time. To determine the fitting tendency, a generalized additive model was used as the regression method. For VPC, 500 simulations were performed on the basis of the model’s fixed and random effects. A normalized prediction distribution error (NPDE) [21] test was additionally conducted to check the normality of the prediction of the two different types of models.

### 2.5. Software for Simulation and Estimation

For the simulation study, R (4.2.2) and its rxode2 package [22] were used. Parameter optimization for the models was performed using NONMEM (7.5.0) and PsN (5.2.6; Perl Speaks NONMEM) software. During the model estimation process, to prevent local minimization, saddle point reset [23] and parameter perturbation were used to confirm the minima state of the parameter set. The estimation method was first-order conditional estimation with interaction (FOCEI).

## 3. Results

### 3.1. Simulation Study

In the simulation in which the one-compartment model’s absorption rate (Ka) was modified to a fractionally modified rate, when the heterogeneity exponent h value was in the range of 0.75–0, the lower peak concentration appeared earlier. When the h value was in the range of 1–0.75, the concentration increased, and the time point to the peak was reached near 0. By contrast, in the case in which the elimination rate (Ke) was adjusted, a larger h resulted in a faster decrease in elimination rate. As a result, the drug amount in the central compartment became trapped, and the concentration stopped moving from a certain point determined by h and its corresponding rate constant value. Changes in PK patterns were more pronounced when the fractionally modified parameter was slower than the other rate constants (Figure 2).

In the two-compartment model simulation, after the fractal expression had been applied, the pattern changes in absorption and elimination were similar to those of the one-compartment cases. When the volume of the peripheral compartment was larger than the volume of the central compartment, the PK pattern did not show a larger difference. When the fractal rate was applied to Kcp, there was little change in the PK profile if the central distribution volume was larger than the peripheral distribution volume. By contrast, the effect of the fractal Kcp was large, and double peaks in the profiles were observed when the rate of absorption was higher than the clearance. In the case of fractal Kpc, the increase in h resulted in a more distinct PK pattern between the distribution and elimination phases (Figure 3).

### 3.2. Numerical Model Evaluations in a Real Case

The fractal rate of the absorption process was applied to the collected models, and estimations in each model were conducted. The minimized state of the model was confirmed on the basis of parameter perturbation and saddle point resets. Performance was measured using OFV, AIC, and AICc. For the OFV, all metrics decreased significantly, but not for Model Case 3. The decrease was greatest in Model Case 2. For AIC and AICc, positive values were obtained only for Model Case 3, which indicated that the addition of the parameter did not enhance model prediction. The decrease in AICc was largest in Model Case 2, followed by Model Case 1. The performance results are summarized in Table 2.

### 3.3. Visual Model Evaluations

In the GOF plots, observations vs. predictions, predictions vs. individual weighted residuals, and time vs. conditional weighted residuals showed reasonable fittings for all model cases. Some prediction discrepancies were observed in Model Case 3, which lacked a structural part. The other models’ fitting points were closely located near the standard lines of y = x and y = 0 in each of the predictions and residuals. Most of the conditional weighted residual values were included within ±2, and the trends lay around zero. With the exception of Model Case 4, the fractal models’ regression lines tended to be closer to the standard lines, and the conditional weighted residuals’ tendency was notably improved by the application of fractal kinetics (Figure 4).

Prediction-corrected VPC plots reveal that all of the models reasonably explained the respective observation data. In the absorption phases that included fractal kinetics, the observations were mostly better covered by the middle range of the prediction intervals. In Model Cases 1 and 5, the NPDEs of the fractal models were more normally distributed than in the models without fractal kinetics. The other model cases showed similar distribution patterns. When combined with the numerical NPDE results, the prediction error tendency of the models was more satisfactory for conditions of normal distribution. Prediction-corrected VPC and NPDE results are shown in Figure 5 and Figure 6.

## 4. Discussion

The simulation study for the one-compartment model shows that a heterogeneity exponent h close to 1 for the absorption process generated a PK profile similar to that of intravenous injection. An h value close to 1 produced a very high rate constant at time points close to 0, implying that the dosed compartment emptied almost instantaneously. The fractal expression of elimination implies that the drug amount remained in the central compartment, with the time when the amount became completely trapped being earlier as the value of h increased. The fractal equation had a significant effect on relatively slow rate constants, implying a larger difference in the distribution of the drug particles between two compartments. These results are consistent with the theoretical underpinnings of fractal concepts.

For the simulation results of the two-compartment model, the fractal absorption and elimination patterns were similar to those of the one-compartment model, in which the volume of the peripheral compartment was smaller than that of the central compartment. Fractal Kcp, which describes the drug’s rate of movement from the central to the peripheral compartment, did not have a large effect on the drug concentration in the central compartment because the peripheral compartment’s capacity to retain the drug was smaller than that of the central compartment. However, when the drug’s estimated peripheral volume of distribution was large, drug distribution between the central and peripheral tissues played a major role in drug elimination from the systemic circulation, such that the sensitivity of drug concentration with respect to fractal absorption was less.

When the fractal rate was applied to Kcp, in the case that the volume of drug distribution in the peripheral compartment was large, a unique PK pattern was obtained in which the drug moved into a large peripheral space and slowly returned to the central space, yielding double peaks in the concentration profile. The h value in Kcp determined the amount of the dose that moved into the peripheral compartment by adjusting the rate of Kcp decay over time. This double-peak pattern was most pronounced when the h value was around 0.5 because the portion of dose in the peripheral compartment gradually decreased as h approached 1. This pattern is often seen for sustained-release drugs that are thought to circulate in the lymph nodesor bind to certain tissue, as in Model Case 3. In the reproduction of this behavior in compartment modeling, it is important to establish certain conditions between parameters; for example, when the volume of distribution is larger in the peripheral compartment than that in the central compartment, the absorption rate should be far greater than the clearance, as shown in Figure 3.

In the case of Kpc, the fractal rate had a greater effect when the volume of distribution was smaller in the peripheral compartment than that in the central compartment. The h value determined the amount of dose retained in the peripheral compartment by adjusting the rate of Kpc decay over time. When the volume of distribution was larger in the peripheral compartment than that in the central compartment, the fractal effect was relatively small, as the value of Kpc was much smaller, such that the dosed amount already tended to be retained in the peripheral compartment after the first dosing event.

Among the five model cases, on the basis of the AIC and AICc results, improved performance metrics only failed to occur in Model 3, such that the decrease in OFV was not significant. Among the models, performance improvement was best for Model 2, followed by Models 1, 4, and 5. All GOF plots and regression lines were in reasonable ranges, with good agreement between observations and predictions. The fractal models generally showed equal or better fitting results in the predictions, except for Model 4, and most of the differences between the conventional and fractal models were not notable according to the visual criteria. The effect of fractal absorption for the conditional weighted residuals vs. time was the most distinctive.

According to the VPC results, the confidence intervals of the prediction for the 5th, 50th, (median), and 95th percentiles show that the observations were better for the models with the fractal rate than those of the base models without the fractal rate with respect to absorption. The 95th percentiles were especially improved in terms of coverage, which showed that the interpretation of high dose concentrations was better in models with a fractal rate.

The prediction confidence intervals of the fractal model tended to be lower in Model Cases 1, 2, and 5 than that in the base model. This implies that the fractional expression can improve interindividual model misspecification, by narrowing the width of the confidence interval, if the misspecification did not originate from uncontrolled variance of the data itself. Although the performance metrics were improved, the confidence intervals could increase, as they better reflect the variance in the observations. The fractal rate can improve the η-shrinkage [24], a measure of the agreement between the observed and model-specified variances. The normalized prediction distributions of the models were not impaired by fractal rate introduction. In all cases, there were improvements on kurtosis, variance converged to 1, and the model prediction errors were better shaped following the normal distribution.

In summary, the models in this study had already been fully minimized; thus, changes in the visual diagnostics of the fractal model’s fitting appeared to be subtle, but the introduction of the heterogeneity exponent was favorable, resulting in a further significant improvement in the given structure. In the evaluation of the pure effects of the fractal kinetics upon a model, no structural modifications of the model occurred except for the parameters concerning absorption, i.e., the important step in which drug particles penetrate largely heterogeneous interfaces. If the model compartments and rate constants can be designed to better incorporate fractal kinetic concepts, simpler and more descriptive models are possible. The fractal kinetic models tested in this study are expected to provide different results in terms of the nonlinearity between doses, and different interpretations of the nonlinearity of the doses, which can have a large effect on predicting the first-in-human (FIH) dose or bioequivalence (BE). For example, as noted above, the application of fractal kinetics can change the way in which variabilities in models are assessed, as simulated variability may differ from that in models based on a homogeneous assumption. Tests that are highly dependent on drug variability, such as the BE test, may be greatly affected by this difference in approach.

With the models tested in this study, most of the fractal-kinetics applications were meaningful, with differences in performance gain. In addition to the fractal expression tested here, there are several other expressions that can reflect spatial characteristics. Additional improvements in model performance depend on whether the appropriate fractal kinetics are applied to the appropriate step in the process of ADME.

## 5. Conclusions

The application of simple fractal kinetics to existing models is valid and provides a better description of PK without losing the mechanistic description of a drug.

## Figures and Tables

**Figure 1 pharmaceutics-15-00304-f001:**
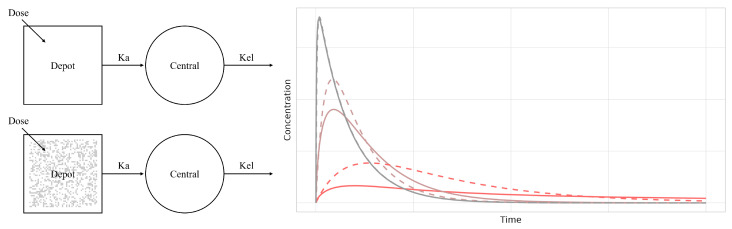
Classical pharmacokinetic model scheme of one-compartment model for oral absorption (upper left scheme, Ka: absorption rate, Kel: elimination rate), pharmacokinetic model with fractal dimension (lower left scheme) and simulation for drug concentration with fractal-like reaction rate to the absorption process (right plot, dashed lines: concentration of classical model; solid lines: concentration of fractal-like rate model with homogeneity exponent of 1/3, red lines: Ka = 0.01, brown lines: Ka = 0.1, gray lines: Ka = 1).

**Figure 2 pharmaceutics-15-00304-f002:**
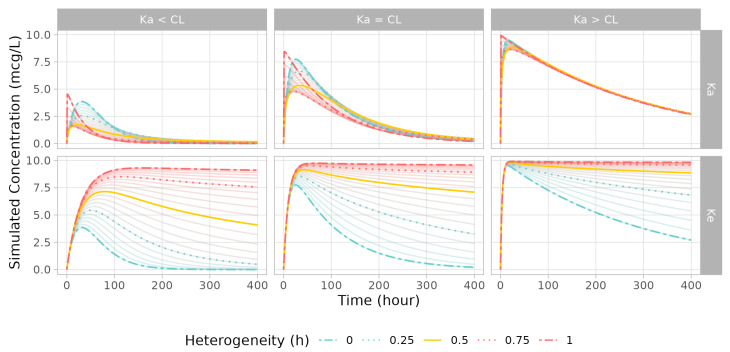
Simulated PK profiles of one-compartment model from parameter sets (upper label) and fractal rate (right label). Changes in the fractal exponent are plotted from 0 (blue line) to 1 (red line) by 0.05. The exponent of 0.25/0.75 is shown with dashed orange lines, and 0.5 is plotted as solid orange line.

**Figure 3 pharmaceutics-15-00304-f003:**
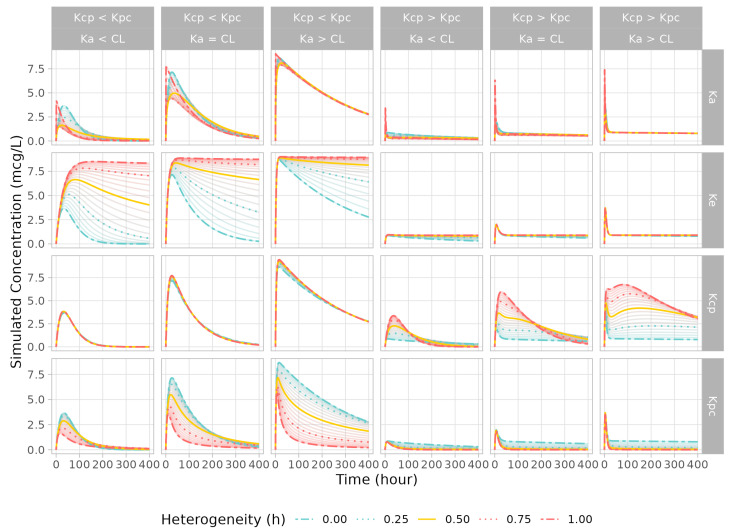
Simulated PK profiles of two-compartment model from parameter sets (upper labels) and fractionally modified rate (right labels). Changes in the fractal exponent are plotted from 0 (blue line) to 1 (red line) by 0.05. The exponent of 0.25/0.75 is shown as dashed orange lines, and 0.5 is plotted as solid orange line. The first three columns present the concentration profile when the central volume is greater than the peripheral volume, and the rest of the columns represent the profiles with the central volume being less than the peripheral volume.

**Figure 4 pharmaceutics-15-00304-f004:**
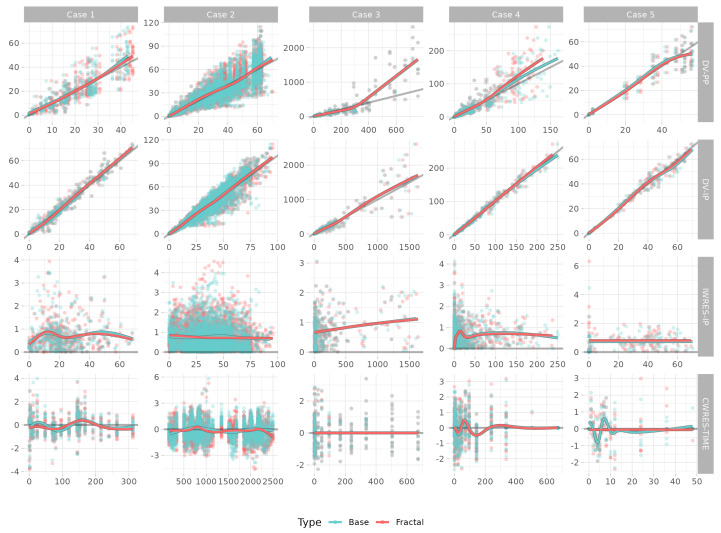
Overlain goodness of fit plots for the fractal and base models. Plots of the x and y axes are displayed in the label on the right side. DV: observation, PP: population prediction, IP: individual predictions, IWRES: individual weighted residuals, CWRES: conditional weighted residuals, TIME: time in hours.

**Figure 5 pharmaceutics-15-00304-f005:**
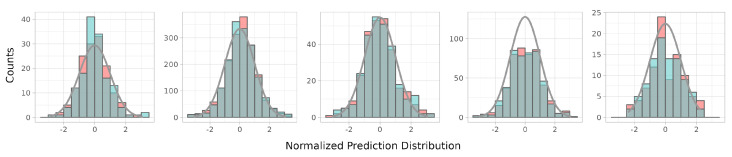
Model performance. Differences are displayed from the mean value of five model outputs.

**Figure 6 pharmaceutics-15-00304-f006:**
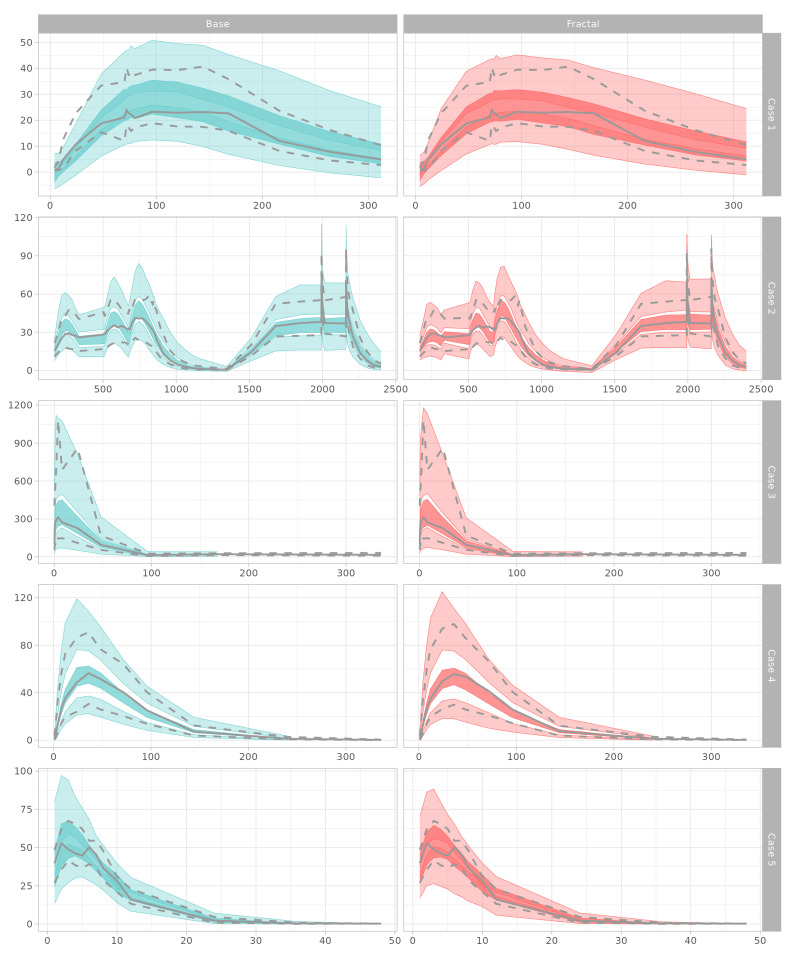
Model performance. Differences are displayed from the mean value of five model outputs.

**Table 1 pharmaceutics-15-00304-t001:** Summary of the conditions and parameters used in the simulation study.

Simulation	A (1-Comp Model)			B (2-Comp Model)			C (2-Comp Model)		
**Condition**	Ka < CL	Ka = CL	Ka > CL	Ka < CL	Ka = CL	Ka > CL	Ka < CL	Ka = CL	Ka > CL
Ka	0.033	0.1	0.3	0.033	0.1	0.3	0.033	0.1	0.3
CL	0.3	0.1	0.033	0.3	0.1	0.033	0.3	0.1	0.033
Q	-			3			3		
V1	10			10			10		
V2	-			100			1		
h	0.00–1.00			0.00–1.00			0.00–1.00		

Ka: absorption rate constant, CL: clearance, Q: intercompartmental clearance, V1: volume of distribution (central),
V2: volume of distribution (peripheral), h: fractal rate constant. Simulations were performed with the conditions
of Ka being larger or smaller than CL, and Ka being equal to CL. Between Simulations B and C, the ratio of central
and peripheral volume of distribution was set to be different.

**Table 2 pharmaceutics-15-00304-t002:** Summary of the performance metrics in the estimation study.

Model	Case 1	Case 2	Case 3	Case 4	Case 5
No. of subjects	18	44	20	40	8
No. of observations	383	3024	339	472	93
No. of parameters—base	12	13	16	21	10
No. of parameters—fractal	13	14	18	22	12
OFV—base	1443.70	13,977.10	2155.43	1556.43	358.47
OFV—fractal	1410.08	13,592.00	2153.54	1539.64	350.13
Δ OFV	−33.62	−385.10	−1.89	−16.79	−8.34
AIC—base	1467.70	14,005.10	2187.43	1598.43	378.47
AIC—fractal	1436.08	13,624.00	2189.54	1583.64	374.13
Δ AIC	−31.62	−381.10	2.11	−14.79	−4.34
AICc—base	1468.54	14,005.24	2189.11	1600.48	381.15
AICc—fractal	1437.07	13,624.18	2191.67	1585.89	378.03
Δ AICc	−31.47	−381.06	2.55	−14.58	−3.12

OFV: objective function value, AIC: Akaike’s information criteria, AICc: corrected Akaike’s information criteria.
Value difference in performance metrics is shown as Δ.

## Data Availability

Pharmacokinetic model schemes of five cases and their parameter estimates of both the base and the fractal models (Appendix A).

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
