# Peer review of "Fractal Kinetic Implementation in Population Pharmacokinetic Modeling"

_pharmaceutics, 2023, doi:10.3390/pharmaceutics15010304_

Round 1

Reviewer 1 Report

The authors make a systematic study of fractal kinetic rates which effectively are power-law time dependent rates. These types of models have been proposed several years ago and sporadically they reappear in literature so a systematic study of their particularities is useful. Furthermore the authors provide 5 examples of their use and claim that in most cases the model is improved with the addition of a fractal rate.

Some issues to consider are the following:

·       It is not entirely clear how the fractal rate of eq. 1 is applied for each case example mathematically. Indicative NONMEM control files would be informative. More specifically I was wondering whether the addition of the term t^-h , affects the units of the original rate constant (θ in Eq. 1)? So do the various “fractal” rate constants have units that depend on h? From the tables in the supplement, it seems that they keep their non-fractal units, but is this correct? If the units do depend on exponent h then this may have important implications when the exponent h is allowed to have IIV as is the case for examples 2, 3 and 5. What is then the meaning of a distribution of a quantity with different units for each of its values?

·       Remove duplicate broken sentence in second line of abstract.

·       Line 275: η-shrinkage, is not “a measure of the agreement between the observed variance and the model-specified variance” as mentioned, it is a measure of a statistical phenomenon observed when the data per individual do not contain enough information to estimate accurately the empirical Bayes estimates and therefore the latter become biased towards the population means and are unreliable.

Reviewer 2 Report

Jung et al describe an analysis of models that include a “fractal” component in the model, basically a time-varying component to give more flexibility to fit observed data. Overall, the manuscript presents novel and interesting work, but does suffer from some drawbacks:

  • Until reading this manuscript, I was unfamiliar with the concept of fractal kinetics (although of course not unfamiliar with fractals, and fractal geometry). I assume many readers will also be somewhat unfamiliar with these concepts. In essence what is described in the manuscript with “fractal” is just introducing a time-varying component in one (or perhaps several) of the model parameters. I think it would be helpful to point that out more clearly in the introduction, perhaps with a very simple example, e.g. just showing a PK curve with fractal-Ka values of 0.2, 0.5, and 0.8. This will help the reader grasp the concept more easily. Also showing how a base ODE would be changed into an ODE with “fractal” absorption would be useful.
  • Also, in general, I think the Introduction can be made a little easier to read. I had to read the introduction again after reading the whole manuscript to start to understand the physiological connection better. For example line 46 it is unclear what is meant with “Projectile”. Can the authors explain that in the manuscript, or use different wording? Would suggest critically rephrasing parts of the Introduction.
  • line 89: “fractal expressions were applied in terms of ADME”. Please add the actual equations that were eventually used in the simulations. Also provide the equations as simulation code in an appendix.
  • section 2.3 and in general: there is a lack of detailed information on how models were implemented. The authors should include documentation of how the models (both simulation and real data) were implemented exactly in software.
  • figure 1: should mention what the labels on the right mean, I suppose this is “the fractionally modified parameter“?
  • line 84: “the value of time … at every point of new dosing”. Whenever a new dose administration happens, is the “time” also reset (and thus the “rate”) for the previous administrations for which absorption might still be ongoing? If so, isn’t this a drawback of this fractal method? That means it’s not truly physiologic anymore? Or, if this is not the case, do you keep track of the rate parameters for different dose administrations in your code? In either case a code example would be useful.
  • By just reading the text and looking at the plots I am not yet convinced why / if the fractal approach is really that beneficial. Maybe the authors can provide more detailed plots that show where the benefit really show? Perhaps plots of individual fits, for example in cases of double-peak absorption?

Round 2

Reviewer 1 Report

I don’t think the authors have answered to my comment in “point 1”.

The authors made no clarifications in the supplementary information to provide the exact mathematical relationships for each example, and did not provide the NONMEM code that would allow the reader to find out what’s going on, unless I have missed it. They only changed their terminology of “rate constant” to “instantaneous rate coefficient” and said that “h” is unitless. That’s all fine. But the issue I raised was what are the units of the “rate coefficient”, are they really inverse-of-time as they appear in tables? Because then it is not clear how the units in ODEs work out. Furthermore if the units are not inverse-time and are instead dependent on “h” (such as time^(h-1), but I am not sure how the authors have formulated it exactly since neither the ODEs or the NONMEM code is provided), then one important implication is how do you consider the population mean of such a quantity when each of the value has a different “h” (since h is allowed to vary between subjects!

Please clarify these issues.

Author Response

Please refer to the attached file thank you. Best regards.

Round 3

Reviewer 1 Report

I think that the key assumption here is that the term t^(-h) is considered to be dimensionless. I think that the authors should add this statement clearly in the text because it is not straightforward. I personally think that this assumption is mathematically questionable because as time, t, clearly has some units, let these be minutes, hours, days or what have you, surely the term t^(-h) is impacted by this choice. I will not insist on this issue any further, because it is theoretical.

Author Response

Please refer to the attached response file (MS Word document). Thank you!
